# Cancer Classification Utilizing Voting Classifier with Ensemble Feature Selection Method and Transcriptomic Data

**DOI:** 10.3390/genes14091802

**Published:** 2023-09-14

**Authors:** Rabea Khatun, Maksuda Akter, Md. Manowarul Islam, Md. Ashraf Uddin, Md. Alamin Talukder, Joarder Kamruzzaman, AKM Azad, Bikash Kumar Paul, Muhammad Ali Abdulllah Almoyad, Sunil Aryal, Mohammad Ali Moni

**Affiliations:** 1Department of Computer Science and Engineering, Green University of Bangladesh, Dhaka 1207, Bangladesh; rabea@cse.green.edu.bd; 2Department of Computer Science and Engineering, Jagannath University, Dhaka 1100, Bangladesh; maksudaoni6@gmail.com (M.A.); alamintalukder.cse.jnu@gmail.com (M.A.T.); 3School of Information Technology, Deakin University, Waurn Ponds Campus, Geelong, VIC 3125, Australia; ashraf.uddin@deakin.edu.au (M.A.U.); sunil.aryal@deakin.edu.au (S.A.); 4Centre for Smart Analytics, Federation University Australia, Ballarat, VIC 3842, Australia; joarder.kamruzzaman@federation.edu.au; 5Department of Mathematics and Statistics, College of Science, Imam Mohammad Ibn Saud Islamic University (IMSIU), Riyadh 11564, Saudi Arabia; kazad@imamu.edu.sa; 6Department of Information and Communication Technology, Mawlana Bhashani Science and Technology University, Tangail 1902, Bangladesh; bikash@mbstu.ac.bd; 7Department of Software Engineering, Daffodil International University (DIU), Dhaka 1342, Bangladesh; 8Department of Basic Medical Sciences, College of Applied Medical Sciences in Khamis Mushyt King Khalid University, Abha 61412, Saudi Arabia; maabdulllah@kku.edu.sa; 9Artificial Intelligence & Data Science, School of Health and Rehabilitation Sciences, Faculty of Health and Behavioural Sciences, The University of Queensland, St Lucia, QLD 4072, Australia

**Keywords:** cancer detection, machine learning, gene data, feature selection, voting classifier, gene analysis

## Abstract

Biomarker-based cancer identification and classification tools are widely used in bioinformatics and machine learning fields. However, the high dimensionality of microarray gene expression data poses a challenge for identifying important genes in cancer diagnosis. Many feature selection algorithms optimize cancer diagnosis by selecting optimal features. This article proposes an ensemble rank-based feature selection method (EFSM) and an ensemble weighted average voting classifier (VT) to overcome this challenge. The EFSM uses a ranking method that aggregates features from individual selection methods to efficiently discover the most relevant and useful features. The VT combines support vector machine, k-nearest neighbor, and decision tree algorithms to create an ensemble model. The proposed method was tested on three benchmark datasets and compared to existing built-in ensemble models. The results show that our model achieved higher accuracy, with 100% for leukaemia, 94.74% for colon cancer, and 94.34% for the 11-tumor dataset. This study concludes by identifying a subset of the most important cancer-causing genes and demonstrating their significance compared to the original data. The proposed approach surpasses existing strategies in accuracy and stability, significantly impacting the development of ML-based gene analysis. It detects vital genes with higher precision and stability than other existing methods.

## 1. Introduction

Cancer, a fatal disease caused by various metabolic abnormalities and inherited diseases, is one of the leading causes of death [1,2,3]. WHO reported that cancer is the first or second leading cause of death before age 70 in 112 of 183 countries and the third or fourth in 23 others [4]. For the majority of the prevalent kinds of cancer, there is a lack of effective medical treatment [5]. The microscopic examination of a tissue sample is a common conventional method for diagnosing cancer, but it is time-consuming, expensive, and occasionally yields unreliable results [6,7]. Other conventional methods rely on the anatomical existence of tumors or factors derived from clinical examinations, but they may produce results that are not precise [8]. Finding the differences among tumor cells requires very highly skilled expertise. This process can be tedious, time-consuming, and quite expensive. These limitations of the general framework lead to the development of additional tumor classification criteria. The recent advancement of the microarray era has inspired the use of gene expression data to analyze genes and detect cancers simultaneously [9]. In gene expression profiles, using microarray facts mixed with computation approach evaluation is considered the most recent technique and process for reliable cancer capability research and can predict more accurate outcomes [10]. From a single data point, microarray operations generate much gene expression data [11]. The abundance of genetic data now makes it possible to track the expression patterns of thousands of genes at once in various experimental settings. In addition, by handling all of the gene information at once, we can analyze it efficiently and accurately [12]. This allows for quick detection and precise therapy, guaranteeing effectiveness and minimizing adverse medication effects [13]. Yet, the high dimensionality and relatively small sample sizes are significant issues of microarray data [14]. Additionally, most microarray cancer data are noisy and might not be particularly helpful in the cancer identification process [15]. Today’s fundamental research problem is identifying the most important cancer-related genes and categorizing cancer types more accurately and reliably [16,17]. The chosen genes aid in illness comprehension, improve cancer classification performance, and lower the cost of medical diagnostics [18]. Hence, to reduce the dimension and redundancy of gene expression data during the classification process using ML, feature selection (FS) is a crucial step. An effective and reliable feature selection approach accelerates classifier learning and overall detection performance.

Many studies [19,20,21] used various data mining, statistical, and machine-learning-based methodologies to examine and analyze the cancer classification challenge since gene expression data typically include a large number of genes. As a result, ML algorithms used to classify cancer have become widespread, attracting researchers’ attention as an emerging approach. Machine learning refers to designing and implementing models and algorithms that enable a system to acquire more knowledge-based intelligence based on prior experience or train data [22,23,24]. For example, machine learning’s standard classification approach involves teaching a classifier to identify patterns from training data and using the trained classifier to categorize unseen samples.

Ensemble techniques merge two or more well-researched strategies to formulate a new approach to obtain a better predictive performance [25,26,27]. Ensemble learning has been used in numerous recent research studies to tackle various data mining tasks, including outlier identification and classification, and to devise ensemble-based feature selection techniques. Among previous studies, the MIMAGA-Selection method was introduced in [25] as an ensemble FS approach that combines mutual information maximization (MIM) and an adaptive genetic algorithm (AGA). The MIMAGA-Selection technique reduced the leukaemia datasets’ dimension and eliminated redundant data. The authors [25] evaluated the performance of the MIMAGA-Selection algorithm concerning classification accuracy using the leukaemia dataset. In addition, they compared the accuracy (97.62% ) of their feature selection approach with other existing methods, including a backpropagation neural network, SVM, extreme learning machine (ELM), and regularized ELM. Likewise, Akadi et al. [28] ensembled minimum redundancy maximum relevance (mRMR) and a genetic algorithm to explore genes related to colon cancer. They applied naive Bayes classifiers and SVM to validate this method, and the model achieved 66.13% accuracy. Salem et al. [29] used a different feature selection method, combining information gain and a genetic algorithm. Genetic programming was then applied to the colon cancer dataset for analysis and gained 85.48% accuracy. Several methods have been proposed in previous studies that have yielded positive results. However, for example, when limiting the number of selected features to a specific number of genes prior to building the classifier, the majority of literature claimed good results, ignoring the significant number of remaining genes that a single selector algorithm, such as PCA, recursive feature elimination, Pearson correlation, or ridge regression, could have neglected.

Many studies claimed that fewer genes would improve classification accuracy. However, the accuracy level still needs improvement. Therefore, an effective ensemble approach is required to be devised, which might enhance the performance of a classification model. To the best of our knowledge, no prior study has applied the ensemble feature selection method using a ranking mechanism based on the frequency of features.

To address the aforementioned issues and improve the accuracy of cancer detection, we focus on selecting essential features utilizing multiple algorithms rather than a single algorithm. Firstly, we proposed a rank-based ensemble feature selection method that extracts significant features from preprocessed data using different feature selection approaches, including FSMs: PCA, recursive feature elimination, Pearson correlation, ridge regression, and variance threshold. The proposed rank-based ensemble feature selection method aggregates features provided by individual FSMs and then ranks them based on the frequency of occurrences of a feature being selected by those FSMs. Secondly, a voting classifier was proposed to achieve higher accuracy using different ML classifiers, such as KNN, DT, and SVM. Extensive experiments were conducted to show the proposed method’s effectiveness. The results of our method were compared with those of the existing single ML algorithms and several built-in ensemble classifiers. Compared to raw data and individual feature selection methods, the experimental results demonstrate that the proposed voting classifier gives a more accurate prediction by selecting the features provided by the rank-based ensemble feature selection approach. In summary, the key contributions to this study are listed below:We proposed an effective rank-based ensemble feature selection approach as well as an ensemble voting classifier model that can perform cancer classification with substantially high accuracy.The rank-based ensemble feature selection approach selects the most influential and relevant features that improve the cancer detection performance by utilizing several feature selection techniques, namely PCA, recursive feature elimination, Pearson correlation, ridge regression, and variance threshold.The ensemble classifier model was developed using an average weighted voting classifier on several ML algorithms, including SVM, KNN, and DT, to enhance the effectiveness of our model.Finally, the model was evaluated with three prominent cancer datasets. The performance results show that our proposed method outperforms the existing works with the selected features.

The remainder of the paper is arranged as follows: Section 2 provides a description of microarray data and a review of the literature on ML-based cancer categorization. Section 3 describes the proposed methodology, feature selection methods, and classification algorithms adopted in the proposal. Section 4 presents the findings of the experiments as well as a discussion. Finally, Section 6 concludes our research, its limitations, and future efforts.

## 2. Preliminaries

As DNA microarray data are often quite large, it is vital to analyze them as soon as possible. Clustering, gene recognition, classification, and gene regulatory network modeling are a few of the most important applications of DNA microarray data analysis. The researchers utilized several machine learning and data mining approaches to implement these. The challenge of gene identification has been solved using information theory [30]. The gene regulatory network modeling challenge has also been approached using Boolean networks [31], Bayesian networks [32], and reverse engineering methods [33]. Fisher linear discriminant analysis [34], decision tree, K-nearest neighbor, multi-layer perceptron [35,36,37], support vector machine [38,39], boosting, and self-organizing maps [40] are only a few examples of ML techniques that have been employed in the past to classify gene expression data. Several data mining techniques, such as ensemble classification [26], outlier identification [41], and a few ensemble-based feature selection approaches [27], have used ensemble learning in recent years. Next, a range of related works are discussed below that employed various techniques for feature selection from the gene expression data.

### Related Works

Machine learning is a popular topic for classifying cancer using microarray data. Numerous algorithms and models are proposed and analyzed by researchers.

Khosro Rezaee et al. [42] proposed a methodology that combined the signal-to-noise ratio, the Wilcoxon technique, and receiver operating characteristic in the feature selection strategy to choose wrapper genes and rank them using the k-nearest neighbor algorithm. Lymphoma, leukemia, and prostate cancer datasets were classified using stacked deep neural networks and soft ensembling to find the most efficient gene subsets.

Matías Gabriel Rojas et al. [43] introduced two memetic micro-genetic algorithms. To prevent altering the algorithm’s direction, the first strategy presents a new local search method that affects fewer of the individuals’ genes. The second method adds a new local search operator to execute a substantial variation in the structure of the exploited neighborhoods, leading to a considerable perturbation over the chosen features of each solution. Three classifiers, i.e., support vector machine, decision tree, and k-nearest neighbors, are used for the classification of seven gene expression datasets.

Hongyu Pan et al. [44] reduced the search space for intricate feature selection issues by using the ReliefF algorithm and Copula entropy as a prefiltering technique to rank feature relevance at the beginning. A modified grey wolf optimization (GWO) algorithm was proposed for feature selection, where a differential evolution algorithm was used to expand the search space of standard GWO.

Amit Bhola et al. [9] gave an overview of numerous cancer categorization approaches and compared and contrasted them in terms of classification precision, computing time, and capacity to uncover knowledge about genomes. This comparison study used a total of seven different categorization algorithms. Similarly, Sung-Bae Cho et al. [12] used three datasets, i.e., leukemia, colon, and lymphoma cancer datasets, to rigorously assess the performance of FSMs and ML classifiers, aiming to examine various features and classifiers.

AliReza Hajieskandar et al. [45] proposed the grey wolf algorithm, which was used in the preprocessing phase to extract notable features, and deep learning was utilized to improve the accuracy of detection of cancer using a deep neural network from three datasets: LUAD (lung adenocarcinoma), STAD (stomach adenocarcinoma), and BRCA (breast invasive carcinoma).

Lu Huijuan et al. [25] presented the MIMAGA-Selection method, which is a hybrid FS method that combines mutual information maximization (MIM) and the adaptive genetic algorithm (AGA). As compared with existing FS techniques, the MIMAGA-Selection algorithm’s utility is proven by its classification accuracy rates. Similarly, Nimrita Koul et al. [46] employed an ensemble FS approach to pick genes’ subsets from the colon cancer dataset. They chose the best five, ten, twenty, and thirty genes using MI as the initial level of gene classification and, as the second level of FS, kernel PCA. An RF classifier with four depth was used to classify the data.

Wei Luo et al. [47] employed SVM to classify cancer. They developed a two-step feature selection strategy that was mixed. To choose discriminatory traits, the first stage employs a modified t-test approach. The second stage uses a modified t-test approach to extract the main elements from the highest-ranking genes.

Murad Al-RajabI et al. [10] presented a framework for a two-stage multifilter hybrid feature selection model for colon cancer classification. The model uses a combination of information gain and a genetic algorithm to deal with feature selection. the next step is to use the minimal redundancy maximum relevance (mRMR) strategy to filter and rank the genes. The data are further examined using correlated ML methods in the last phase. It was found that DT, KNN, and NB classifiers showed promisingly accurate results using the developed hybrid framework model.

Harikumar Rajaguru et al. [48] employed two ML algorithms, i.e., DT and KNN, for the categorization of breast tumours. The Wisconsin Diagnostic Breast Cancer (WDBC) dataset is used to verify these two machine learning techniques following feature selection using principal component analysis. The two ML algorithms are compared using common performance measures. The KNN classifier performs better than the DT classifier in the classification of breast cancer according to the results of the comparison investigation.

In [49], Kesav Kancherl et al. suggested a model that made use of the recursive feature elimination (RFE) approach, which is based on the support vector machine (SVM). They achieved 87.5% accuracy.

Ashok-Kumar Dwived et al. [50] used an artificial neural network (ANN). The ANN was also contrasted with five other ML methods. This study reports a significant classification accuracy of 98% utilizing the ANN, with no error in the identification of acute lymphoblastic leukemia and just one error in the identification of acute myeloid leukemia using a ten-fold cross-validation and leave-one-out technique.

Hajar Saoud et al. [51] evaluated how well the Bayes network (BN), SVM, KNN, ANN, DTC, and logistic regression performed in the diagnosis of breast cancer in order to determine whether this particular cancer is a benign or malignant tumor. The algorithms are simulated with the WEKA tool using the Wisconsin breast cancer dataset from the UCI machine learning repository. With an accuracy of 97.28%, the Bayes network and support vector machine (SVM) algorithms produced the best results.

Reinel Tabares-Soto et al. [13] presented an empirical study that encompasses different kinds of machine learning and deep learning algorithms. To categorize the tumors, they contrasted commonly used methods in standard machine learning and deep learning. They used the “11-tumor database”. For feature selection, PCA was utilized. Applying PCA as a feature selection method and logistic regression as a classifier, they were able to accurately identify tumors 94.29% of the time.

Table 1 presents different approaches to applying feature selection methods and classifiers.

## 3. Methodology

The main principle of our study is depicted as a schematic diagram in Figure 1. In summary, the proposed methodologies comprise the following steps.
For the preprocessing of three experimental cancer datasets that were used in this study, namely leukemia, colon, and the 11-tumor dataset, unnecessary columns, missing values, and duplicate rows were removed, followed by label encoding and normalization.Next, significant and relevant features were extracted from the preprocessed data using different feature selection methods (FSMs), such as PCA, recursive feature elimination, Pearson correlation, ridge regression, and variance threshold, as well as our suggested rank-based ensemble feature selection method, by integrating multiple feature selection approaches.Then, the data split was conducted with a 70:30 ratio between the train and test datasets, with the training dataset serving as a calibration dataset for the model’s parameters and the test dataset serving as an evaluation dataset for performance.Then, the reduced dataset was evaluated using appropriate classification assessment metrics for a variety of ML classifiers, including KNN, DT, SVM, and proposed voting classifiers.We further compared the proposed voting classifier with the built-in ensemble classifiers such as AdaBoost (AB), gradient boost (GB), and random forest (RF).

In the following subsections, the proposed methodologies are described in detail.

### 3.1. Data Acquisition and Preprocessing

Several microarray datasets from cancer gene expression research that have been released are available, including leukemia, colon, prostate, breast, 11-tumor, lymphoma, and lung cancer dataset, etc. Among them, we used 3 different cancer datasets in our study, i.e., leukemia cancer, colon cancer, and the 11-tumor dataset. These benchmark datasets have been used in several earlier studies and contain high-dimensional data since the number of characteristics exceeds the number of samples [13]. After preprocessing the data, we divided them into 70:30 train and test datasets, with the training dataset used to calibrate the model’s parameters and the testing dataset used to evaluate performance. Splitting comprises 50 samples for training and 22 samples for testing in the leukemia dataset. Splitting comprises 43 training samples and 19 testing samples for the colon dataset, and 121 training samples and 53 testing samples for the 11-tumor dataset.

Preprocessing is a crucial exploratory step for any analytical tasks, as it may be inherently noisy. For this study, we conducted the preprocessing steps on the datasets, including deleting extraneous columns, checking for missing values and duplicate rows, and encoding labels in our experimental dataset. Some ML algorithms, such as SVM and KNN, rely on the distance between the observations for accurate classification. Normalizing the training data can enhance their performance considerably if the features represent distinct physical units or come in wildly different scales. As a result, we have to preprocess the data using scaling, which ensures that the values are within a reasonable range.

### 3.2. Proposed Rank-Based Ensemble Feature Selection Approach

Feature selection (FS) is a method for selecting the most important properties in any predictive analysis to improve its performance. The primary goals of FS are to improve predictive accuracy, remove unnecessary features, and minimize the time spent analyzing the data. In that process, FS identifies the best subset of features that can be used to create effective models of the phenomenon being examined. Thus, selecting relevant genes from microarray data improves the accuracy of the cancer classification process. In this study, we introduced a rank-based ensemble feature selection method that selects the most relevant feature from different feature selection algorithms as shown in Figure 1.

We employed a variety of different feature selection methods, namely PCA, recursive feature elimination (RFE), Pearson correlation (PC), ridge regression (RR), and variance threshold (VRT), within the same dataset to select different subsets of the existing features. Then, we aggregated the different subspace features selected by each of the selectors. Then, the aggregated features were sorted according to their individual rankings. In any ensemble scheme, the combination of partial results into a final output is a critical step in determining success. It is common practice to combine the various features selected by the various selectors. In our study, we applied three different approaches to select the N features from all the aggregated features as follows:Ensemble-1 (E1): Select aggregated features with a frequency of occurrence greater than one. That is the feature selected by at least two FS methods.Ensemble-2 (E2): Select aggregated features with a frequency of occurrence greater than two. Thus, the feature is selected by at least three FS methods.Ensemble-3 (E3): In the third approach, features selected by more than three FS methods are chosen.

Algorithm 1 represents the overall procedure of the feature ensemble process. Initially, the sub-feature sets $S_{1}$, $S_{2}$, $S_{3}$ are set to 0. Then, for each of the selected features in the aggregate feature, a rank is assigned. The number of feature selectors that select a particular feature $f_{i}$ is set as the rank $R_{i}$ of that feature.
**Algorithm 1** Rank-based ensemble feature selection
Initialize each sub-feature set: $S_{1}=0,S_{2}=0,S_{3}=0$.   1:**for** each feature $f_{i}\in \left\{1,\ldots ,i\right\}$ **do**   2:      **for** each Feature Selector ${FS}_{j}\in \left\{1,\ldots ,j\right\}$ **do**   3:             **if** $f_{i}==Selected$ **then**   4:                   $R_{i}\leftarrow R_{i}+1$   5:             **end if**   6:      **end for**   7:**end for**   8:**for** each feature $f_{i}\in \left\{1,\ldots ,i\right\}$ **do**   9:      **if** $R_{i}\geq 1$ **then** 10:            $S_{1}\leftarrow S_{1}\cup f_{i}$ 11:      **end if** 12:      **if** $R_{i}\geq 2$ **then** 13:            $S_{2}\leftarrow S_{2}\cup f_{i}$ 14:      **end if** 15:      **if** $R_{i}\geq 3$ **then** 16:            $S_{3}\leftarrow S_{3}\cup f_{i}$ 17:      **end if** 18:**end for**


This ranking process is depicted in lines 1–7 of Algorithm 1. Each of these features is added to the sub-feature space $S_{1}$, $S_{2}$, or $S_{3}$, depending on the importance of the features. As the subset $S_{1}$ contains the features that have been selected by at least one FS, the total number of features in this subset exceeds that of any other. On the other hand, the subset $S_{3}$ contains a lower number of features. All these steps are described in lines 8–18.

### 3.3. Proposed Ensemble Voting Classifier for Cancer Detection

The ensemble voting classifier is an ML technique that learns from a group of models and predicts an output class based on the output’s highest chance of being the desired class. This classifier sums up the outcomes of each predictor that has been fed into the voting classifier and guesses the output class with the most votes. As a second contribution, rather than building multiple specialized models and evaluating their effectiveness, we created a single ensemble classification model based on the ML classifiers, namely SVM, KNN, and DT. This ensemble model is trained on the selected feature space described in the previous subsection and evaluates its performance. Usually, the voting classifier can predict the classification results based on the total number of votes cast for each predicted output. In our ensemble model, we employed a weighted average (hard) voting classifier that is based on the different weights of each classifier and combined the classification results of the base classifiers to increase the performance and make a reliable prediction model. The weights that we utilized in our ensemble models are 1, 2, and 3 based on the accuracy score of SVM, KNN, and DT algorithms.

The voting ensemble classifier’s overall process is represented by Algorithm 2. Initially, L is set to label tuples of each class with selected features, D is set to tuples for evaluation, T is set to tuples for evaluation, and N is set to ML classifier. Then, for each ML classifier, models are trained using labeled data. This process is depicted in lines 1–3. Then, for each ML classifier, models are tested using testing data. This process is depicted in lines 4–6. Then, a weight is assigned to each ML classifier based on its accuracy value. This process is depicted in lines 7–9. Eventually, for each set of test data, the predicted class is found using each ML classifier and votes for classifiers are taken based on the weights assigned to classifiers. At last, the output class is guessed based on the most votes. All these steps are depicted in lines 10–15.
**Algorithm 2** Majority voting ensemble classifier
Input:
- L: Labeled tuples for training of each class C with selected features.
- D: Set of tuples for evaluation.
- T: Set of tuples for testing.
- N: ($ML_{1}$, $ML_{2}$, $ML_{3}$, …, $ML_{N}$) Set of classifier ML algorithm.
   1:**for** each ML classifier $ML_{i}\in \left\{1,\ldots ,N\right\}$ **do**   2:      Train $ML_{i}$ using labeled data *L*   3:**end for**   4:**for** each ML classifier $ML_{i}\in \left\{1,\ldots ,N\right\}$ **do**   5:      Test and evaluate $ML_{i}$ using testing data *D*   6:**end for**   7:**for** each ML classifier $ML_{i}\in \left\{1,\ldots ,N\right\}$ **do**   8:      Assign weight of $ML_{i}$ based on the accuracy of $ML_{i}$   9:**end for** 10:**for** each each test data $t_{i}\in \left\{T\right\}$ **do** 11:      **for** each ML classifier $ML_{i}\in \left\{1,\ldots ,N\right\}$ **do** 12:            Find the predicted class of $t_{i}$ 13:      **end for** 14:      Aggregate vote to ensemble based on the results 15:**end for**


### 3.4. Adopted Feature Selection Methods

Feature selection is used to either obtain a small number of features to avoid overfitting or to prevent features from being redundant or irrelevant. Additionally, it is beneficial to only include the most pertinent and practical data in machine learning training sets, which significantly lowers expenses and data volume.
**Principal Component Analysis:** In nature, cancer data sets have substantially large dimensions. Hence, the number of features is reduced using the principal component analysis (PCA) technique. PCA is an orthogonal linear transformation that converts data to a new dimension system with the biggest variance of any projection of the data falling on the first dimension (called the first principal component), the second-best variance on the second dimension, etc. This method converts a set of observations of possibly correlated variables into a set of uncorrelated variables known as principal components via an orthogonal transformation [54]. The dimensions of various data sets are reduced using this strategy.**Recursive Feature Elimination (RFE):** RFE is a popular method as it is simple to implement and use and is effective in determining which features (columns) in a train set are more or less helpful in determining the target variable. RFE assesses the features by significance and returns the top-n features after removing the least important ones, where n is the users’ input. To use it, the class with the estimator argument is first set up by specifying the algorithm to use and the n_features_to_select argument by specifying the number of features to select. To pick the features, the class has to be suited to a training dataset using the fit() function after it has been configured. In our study, we used the SVM algorithm as an estimator as well as setting n_features_to_select to 5490 for the leukemia dataset, 1000 for the colon dataset, and 5500 for the 11-tumor dataset.**Pearson Correlation Coefficient:** Pearson correlation and Spearman correlation are both measures of association or correlation between two variables, but they serve different purposes and have different strengths and weaknesses. The Pearson correlation coefficient is preferred when the underlying assumption of the relationship is considered to be linear. However, for the Spearman correlation, this assumption is not mandatory, i.e., the variables could be non-linearly related. As we were interested in finding a linear relationship, we used the former one, i.e., the Pearson correlation. Moreover, the types of data that we were dealing with were continuous, which also suits the Pearson correlation metric, whereas the Spearman correlation metric is suitable for ordinal, interval, or ratio data, which was not the case for us in this study. Pearson correlation shows how closely two variables are related linearly. Features with a strong association are more linearly dependent and hence affect the dependent variable in a similar way. If there is a strong association between two traits, one of them might be removed. Only metric variables are appropriate for PC. A correlation coefficient *r* is a number that ranges from −1 to +1: near 0 means a low association (an exact 0 implies no correlation); closer to 1 indicates a strong positive relationship; and near −1 indicates a strong negative relationship. For feature X having values x and classes C having values c, where X and C are viewed as random variables, Pearson’s linear correlation coefficient is calculated as [55]:
(1)$$r\left(X,C\right)=\frac{\sum_{i=1}^{n}\left(x_{i}-\bar{x_{i}}\right)\left(c_{i}-\bar{c_{i}}\right)}{\sqrt{\sum_{i=1}^{n}{\left(x_{i}-\bar{x_{i}}\right)}^{2}\sum_{j=1}^{n}(x_{j}-\bar{x_{j}}{)}^{2}}}$$In our study, we compared feature correlations and eliminated one of two features with a correlation greater than 0.9 (leukemia and colon dataset) and 0.6 (11-tumor dataset).**Ridge Regression:** Ridge regression is a prominent method for predicting data that makes use of regularization. Overfitting is a problem that regularization aims to solve. When there is a huge data collection with thousands of features and entries, overfitting becomes an apparent problem. When the data contain features that are certain to be more relevant and valuable, RR performs better. When a large number of characteristics are included, it is commonly employed to create parsimonious models. It applies L2 regularization, which entails adding a penalty equal to the square of the coefficients’ magnitude. Thus, RR optimizes the following:
(2)Objective=RSS+alpha×(totalofsquareofcoefficients)Here, RSS refers to the “Residual Sum of Squares”, which is nothing but the total of the square of mistakes between the predicted and actual values in the training data set, and *alpha* is the parameter that balances the degree of significance given to lessening RSS vs. minimizing the total of the square of coefficients, where *alpha* can take various values. In our study, we used *alpha* values of 0.3 (leukemia and 11-tumor datasets) and 0.4 (colon dataset).**Variance Threshold:** A simple baseline technique for selecting features is the variance threshold. It eliminates all features whose variance falls below a threshold value. All zero-variance characteristics are removed by default; that is, characteristics that have the same value across all samples. We feel that characteristics with a greater variance include more important data. To apply the variance threshold feature selection method, a variance threshold value (i.e., 0.1, 0.2 etc.) is chosen. The minimum variance that a feature must have to qualify as informative is determined by this threshold. Choosing the appropriate threshold value is crucial; a higher threshold retains only features with higher variance, while a lower threshold retains more features. The choice should align with the nature of the dataset. In our work, we have used different threshold values for different datasets.

### 3.5. Adopted ML Algorithm for Voting Classifier

We used three different classification algorithms. One of them, and the most popular one, is k-nearest neighbors (KNNs). Another one is the support vector classifier (SVM), which is a linear model that is quick to train, predict, and scale well across two datasets. Also, we used a decision tree (DT) classifier, which is a constant data scaling model, unlike linear approaches. To boost classification performance, we merged these classifiers with a weighted voting classifier.
**K-Nearest neighbors:** For handling classification problems, the k-nearest neighbors (KNNs) technique is a simple supervised ML algorithm. The KNN algorithm believes that similar objects are close together. The KNN algorithm compares the *k* closest data points in the training data set based on resemblance metrics to determine the label of input data for a given new data [56]. In this system, the conclusion is selected by a majority vote of its neighbors. Because it requires zero experience and builds a fresh model for every experiment, this algorithm is one of the most efficient ML algorithms available. If the number of instances in the input data set grows, testing may become expensive. In our research, we only use one parameter in the model creation. n_neighbors is set to 3, which means 3 neighborhood points are required for classifying a given point.**Support Vector Machine:** The support vector machine (SVM) was first presented by [57]. SVM is a supervised learning algorithm that was first used for classification and regression. The SVM creates hyperplanes that can be used for classification to maximize the distance between classes. To put it another way, to make the gap between the two categories as large as feasible, SVM translates training examples into points in space. Following that, new examples are mapped into the same space and categorized according to which side of the gap they land on. The kernel parameter of SVM was set to ‘RBF’ (radial basis function) in our model. The RBF kernel works by mapping the data onto a high-dimensional space by finding the dot products and squares of all the features in the dataset and then performing the classification using the basic idea of linear SVM.**Decision Tree Classifier:** A decision tree is a supervised classifier that can be used to solve regression and classification tasks. In this tree-structured classifier, internal nodes reflect dataset properties, branches reflect decision rules, and each leaf node delivers the judgment. In a decision tree, the process of deciding the class of a given dataset begins at the tree’s root node. This method compares the values of the root property to the values of the record (actual dataset) attribute, continues the branch, and moves to the next node. The method compares the value of the property with the values of the other sub-nodes before moving on to the next node. It continues in this manner until it reaches the leaf node of the tree. The model is built using two DT parameters, including randomstate, which is set to 0, and maximumdepth, which is set to 2.

### 3.6. Built-In Ensemble Methods for Performance Comparison

To demonstrate the efficacy of the suggested voting ensemble classifier, the results of the suggested voting ensemble classifier are compared with the following existing built-in ensemble classifiers:**AdaBoost:** Yoav Freund and Robert Schapire [58] proposed AdaBoost, or adaptive boosting, as an ensemble boosting classifier in 1996. To enhance classification performance, it mixes several classifiers. AdaBoost is an iterative ensemble creation algorithm. By aggregating multiple low-performing learners, the AdaBoost classifier builds a formidable classifier with substantially high precision. The main premise of AdaBoost is to train the sample data and build the strength of the classifiers in each step so that trustworthy predictions of unusual observations may be made. Using two AdaBoost parameters, the model is trained, including *n* estimators, which has a value of 50, and randomstate, which has a value of 0.**Random Forest:** Random forest [59] is an ensemble classifier that comprises several decision trees and outputs a class, which is the mode of the class’s output by individual trees. RFs generate a large number of classification trees without the need for trimming. Each class in each classification tree receives a set number of votes. The algorithm selects the category with the most votes from all of the trees. A random forest is an efficient approach for large datasets. However, it is more time-consuming than other methods. It can effectively estimate missing values, making it useful for datasets with a large number of missing values. With the exception of the Gini impurity, the RF trees used a number of DT parameters at default settings. However, two DT parameters are used to create the model. The maximumdepth is set to 2, and the randomstate is random, which is set to 0.**Gradient Boosting:** Gradient boosting is an ML approach that can be used for regression and classification, among other applications. It provides a forecasting model in terms of a group of poor estimation techniques, the most frequent of which are decision trees [60]. If a decision tree is a poor responder, the resulting strategy is known as gradient-boosted trees, and it generally beats random forest [61]. A gradient-boosted tree model is built in the same way as other boosting methods, but it varies in how it can optimize any differentiable loss function. Like the RF tree, gradient boosting uses a number of DT parameters at default settings. However, two DT parameters are used to create the model, including maximumdepth with a value of 2, and the randomstate is random, with a value of 1.

### 3.7. Evaluation Metrics

We employed different performance matrices to evaluate our hypotheses, which are described below:**Accuracy Score:** Accuracy measures how properly the classifier anticipates the classes [62]. The average number of samples is correctly classified by the classifier. The average fraction of correctly predicted samples out of total samples is
(3)$$Accuracy=\frac{Correctly\;Predicted\;Data}{Total\;Testing\;Data}\times 100\%$$**Confusion Matrix:** The confusion matrix is a technique for assessing performance in the form of a table that incorporates information about both actual and expected classes. It is a two-dimensional matrix, with the rows representing the actual class and the columns representing the predicted class. Figure 2 shows the confusion matrix.For two or more classes, the matrix depicts actual and anticipated values. The explanation of the terms of TP, FP, TN, and FN are:
The total number of correct outcomes or forecasts where the actual class was positive is known as true positive (TP).The total number of incorrect results or forecasts where the actual class was positive is known as false positive (FP).The total number of correct outcomes or predictions where the actual class was negative is known as true negative (TN).The total number of incorrect outcomes or forecasts where the actual class was negative is known as false negative (FN).**AUROC Curve:** The area under receiver operator characteristics (AUROCs) curve is a graphical representation of a binary classifier’s performance across various classification thresholds. It plots the true positive rate (sensitivity) against the false positive rate (specificity) at different threshold settings. The curve’s shape offers a wealth of information, including what we care about most for an issue, the expected false positive rate, and the expected false negative rate. To be precise, lower false positives and higher true negatives are shown by lower values on the x-axis of the plot, and higher values on the y-axis of the figure show higher true positives and lower false negatives.

## 4. Experimental Results Analysis

Nine separate datasets were created for each initial dataset during the FS. No FSM is used in the first dataset; PCA is used in the second dataset; RFE is used in the third dataset; PC is used in the fourth dataset; RR is used in the fifth dataset; VRT is used in the sixth dataset; and an EFSM method is used in the remaining three datasets. Datasets were split into 70 percent training data and 30 percent validation or testing data to train and evaluate the performance of the models in each experiment. Several ML algorithms were employed to compare the performance, including SVM, KNN, DT, and voting classifiers.

### 4.1. Dataset Description

**Leukemia cancer dataset:** The leukemia cancer dataset contains 72 samples with 7132 gene expression microarrays, where 47 samples have acute lymphoblastic leukemia (ALL) and 25 samples have acute myeloid leukemia (AML). This dataset is a binary dataset, where the classes are numbered from 0 to 1, each signifying a different type of cancer. The gene expression measurements were taken from 63 samples of bone marrow and 9 samples of peripheral blood. High-density oligonucleotide microarrays were used to evaluate gene expression levels in these 72 samples [12]. The frequency of the number of classes in our test dataset is shown in Table 2.**11-tumor dataset:** The 11-tumor dataset contains 174 samples with 12,533 gene expression microarrays, where 27 samples have ovary cancer, 8 samples have bladder cancer, 26 samples have breast cancer, 23 samples have colorectal cancer, 12 samples have gastroesophageal cancer, 11 samples have kidney cancer, 7 samples have liver cancer, 27 samples have prostate cancer, 6 samples have pancreas cancer, 14 samples have adenocarcinoma cancer, and 14 samples have lung squamous cell carcinoma cancer. The classes are numbered from 0 to 10, each signifying a different type of cancer. This dataset is a multiclass dataset. The “11-tumor dataset” is freely available online at [13]. The frequency of the number of classes in our test dataset is shown in Table 2.**Colon cancer dataset:** The colon cancer dataset contains 62 samples of colon epithelial cells taken from colon cancer patients with 2000 gene expression microarrays, where 40 samples are colon cancer samples, i.e., abnormal samples, and 22 samples are normal. Although the original data contained 6000 gene expression levels, 4000 were deleted due to the lack of confidence in the measured expression levels [12]. Using high-density oligonucleotide arrays, every sample was obtained from cancerous and normal healthy regions of the colons of the same patients. This dataset is a binary dataset. The classes are numbered from 0 to 1. The frequency of the number of classes in our test dataset is shown in Table 2.

### 4.2. Rank-Based Ensemble Feature Selection Process

The number of features selected by different feature selection methods, such as principal component analysis (PCA), recursive feature elimination (RFE), Pearson correlation coefficients (PCs), ridge regression (RR), variance threshold (VRT), and our suggested rank-based ensemble feature selection method, such as ensemble 1 (feature selected by at least two feature selection methods), ensemble 2 (feature selected by at least three feature selection methods), and ensemble 3 (feature selected by more than three feature selection methods), is presented in the following Table 3.

### 4.3. Experimental Results

As we stated earlier, the leukemia cancer dataset contains 72 samples with 7132 gene expression microarrays, the colon cancer dataset contains 62 samples with 2000 gene expression microarrays, and the 11-tumor cancer dataset contains 174 samples with 12,533 gene expression microarrays. At first, the classification capabilities of different individual feature selection methods and classification algorithms were inspected for the classification of three benchmark datasets. Table 4 and Figure 3 present the results of these inspections. From this, in the leukemia dataset, we find that the voting classifier improves accuracy for RFE, PC, and VRT, while the DT classifier has the highest classification accuracy. In the colon dataset, the voting classifier improves accuracy for RR and VRT, while the SVM classifier has the highest classification accuracy. In the 11-tumor dataset, the voting classifier improves accuracy for RFE, PC, RR, and VRT, while the SVM classifier has the highest classification accuracy.

Secondly, feature subsets selected by the proposed rank-based ensemble feature selection method are classified using different individual classifiers and proposed voting classifiers. Here, the classification accuracy value is treated as the primary objective to demonstrate the effectiveness of the proposed rank-based ensemble feature selection methods. The accuracy value of this experiment is shown in Table 5 and Figure 4. It is observed that the voting classifier is better than others for our proposed ensemble E1 and E2 in the leukemia dataset, E2 and E3 in the colon dataset, and E1, E2, and E3 in the 11-tumor dataset. It can be observed that rank-based ensemble feature selection provides higher results than a single feature selection algorithm for E2 in the leukemia dataset (100%) and for E1 in the 11-tumor dataset (94.34%). It can also be observed that SVM performs worse than other classifiers. Yet, despite being trained on a significantly smaller feature set, our proposed rank-based ensemble feature selection improves accuracy for the SVM classifier by E3 in both leukemia and 11-tumor datasets. For all datasets, we assigned weights of 1, 2, and 3 based on the accuracy score of each ML algorithm.

Eventually, to analyze the performance of the proposed voting classifier using rank-based ensemble feature selection, we compared the proposed voting classifier with built-in ensemble classifiers such as AdaBoost, gradient boosting, and random forest. We analyzed the performance of each classifier using accuracy, precision, recall, and f1-score values. The results of this experiment are presented in Figure 5, Figure 6 and Figure 7. Our proposed voting classifier improves E2 and E3 in the colon dataset and E1, E2, and E3 in the 11-tumor dataset. Hence, the voting classifier outperforms other classifiers. Table 5 also shows that the applied voting classifier provides an accuracy rate of 100% and 97% for E2 in leukemia and colon cancer datasets, respectively, and 94% for E1 in the 11-tumor dataset.

Figure 8 shows the confusion matrix for the best results for all datasets, where we can obtain high true positive and negative rates and low false positive and negative rates. It can be observed that the proposed E3 FSM can predict all ALL and AML samples with no errors in the leukemia dataset, whereas the proposed E2 FSM has just one incorrect classification in the colon dataset and the proposed E1 FSM has just a few incorrect classifications.

Figure 9 shows the AUROC curve of the testing dataset for SVM, KNN, DT, and voting classifiers with the highest performance for the leukaemia, colon, and 11-tumor datasets. Also, it provides us with AUC values for models. The higher the AUC values, the better the model’s performance at distinguishing between the positive and negative classes. The AUC scores for the SVM, KNN, DT, and voting classifier in the leukaemia dataset are 0.50, 0.862, 0.895, and 1.0. In the colon dataset, the AUC scores for the SVM, KNN, DT, and voting classifier are 0.90, 0.90, 0.742, and 0.962, respectively. In the 11-tumor dataset, the AUC scores for the SVM, KNN, DT, and voting classifier are 0.766, 0.836, 0.724, and 0.932, respectively. As the AUC score for the voting classifier is the maximum score in terms of all datasets, it is better than others.

## 5. Discussion

Cancer classification using gene expression data is still an active area of research, where a number of proposed methods have yielded satisfactory results.

Earlier, several research studies suggested procedures to select genes and optimize classification accuracy. Some of these techniques used the memetic micro-genetic [43] and ReliefF algorithm and Ccpula entropy as a prefiltering method and a modified gray wolf optimization algorithm for feature selection [44], while others utilize the signal-to-noise ratio, Wilcoxon method, and receiver operating characteristic to select genes and a stacked deep neural network as a classifier [42]. All of the techniques employ gene selection to enhance classification accuracy. However, the rank-based ensemble technique for feature selection and classification has been an unexplored option.

Our proposed methodology can choose a subset of informative features and provide an accurate prediction using the selected features. It is noteworthy that prior algorithms had already chosen the feature but had omitted considering the bare minimum of genes that were truly beneficial. Our simulation research examined the effectiveness and prediction accuracy of the proposed model under several scenarios. We applied the proposed method to some public gene expression data and the results show that the proposed method can appropriately classify various samples based on gene expression. Finally, we compared the proposed voting classifier with built-in ensemble classifiers to examine their performance. For each classifier, we analyzed the performances using accuracy, precision, recall, and f1-score values. We find that our proposed voting classifier outperforms other classifiers.

## 6. Conclusions

Cancer is one of the major causes of death for living beings, necessitating early detection, diagnosis, and medication to keep the disease under control. ML algorithms are potential tools for detecting cancer and its type from complex datasets like microarrays. This paper presents an ensemble-based methodology for cancer classification based on three publicly available benchmark datasets, i.e., leukemia, colon, and 11-tumor. We examined three ML algorithms and six feature selection approaches, including our proposed methodology. The proposed ensemble feature selection method integrates several feature selection techniques, namely PCA, recursive feature elimination, Pearson correlation, ridge regression, and variance threshold, to find the most relevant features and improve cancer detection performance. Furthermore, the proposed ensemble weighted average voting classifier was built with the help of a combination of SVM, KNN, and DT machine learning algorithms to enhance the effectiveness of our model. The performance of these approaches in terms of widely used classification performance metrics was evaluated. The proposed model (Table 6) performed the best, with a classification accuracy of 100%, 94.74%, and 94.3% in the voting classifier for leukemia, colon, and 11-tumor datasets, respectively. Consequently, the proposed approach has an excellent overall performance for the three datasets compared to the related research, as discussed in the results analysis section. The proposal’s experimental results will help researchers to choose the optimal classification approach for specific bioinformatics challenges. However, the proposed approach has some limitations. We did not take into account any oversampling or undersampling techniques for data balancing. Additionally, we did not tune the parameters of our models and we did not explore sophisticated deep-learning-type models in our research. Therefore, future studies will focus on resolving these limitations, i.e., evaluating the performance of the proposed algorithm while taking into account the oversampling and undersampling strategies, assessing the performance of the proposed algorithm while using different kinds of more robust data, and testing the impact of different parameters on the proposed algorithms.

## Figures and Tables

**Figure 1 genes-14-01802-f001:**
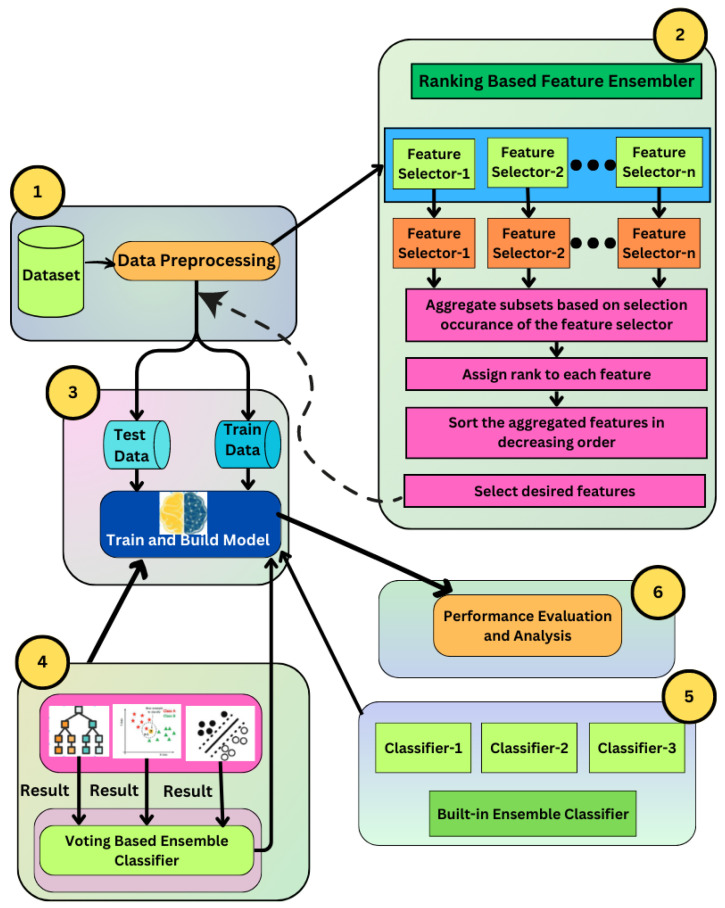
The methodology process is illustrated in a workflow diagram. (1) Preprocessing was performed on three datasets, namely leukaemia, colon, and 11-tumor datasets. (2) Using different FSMs, such as PCA, recursive feature elimination, Pearson correlation, ridge regression, variance threshold, and also proposed rank-based ensemble feature selection, significant features were extracted. (3) Dataset was split into 70:30 train and test datasets. (4) Reduced dataset was trained using ML classifiers, including KNN, DT, SVM, and the proposed voting ensemble classifier. (5) Further voting classifier was compared with built-in ensemble classifiers such as AdaBoost, gradient boost and random forest classifier. (6) Using different performance matrices, such as accuracy and confusion matrix, the performance of the model was assessed and analyzed.

**Figure 2 genes-14-01802-f002:**
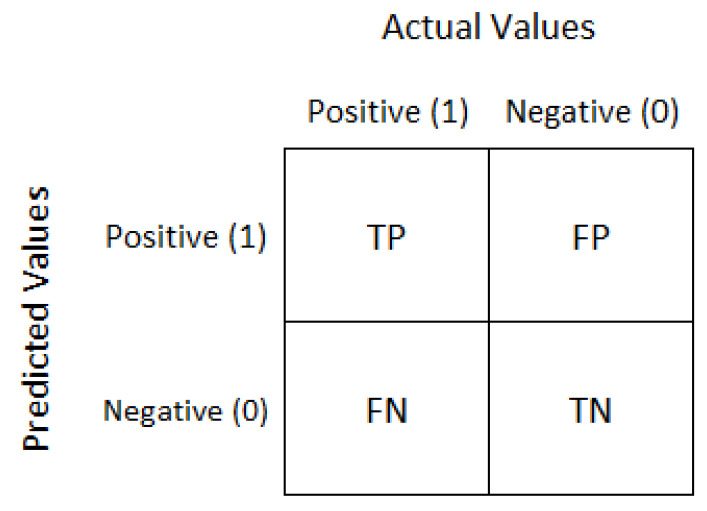
Confusion matrix.

**Figure 3 genes-14-01802-f003:**
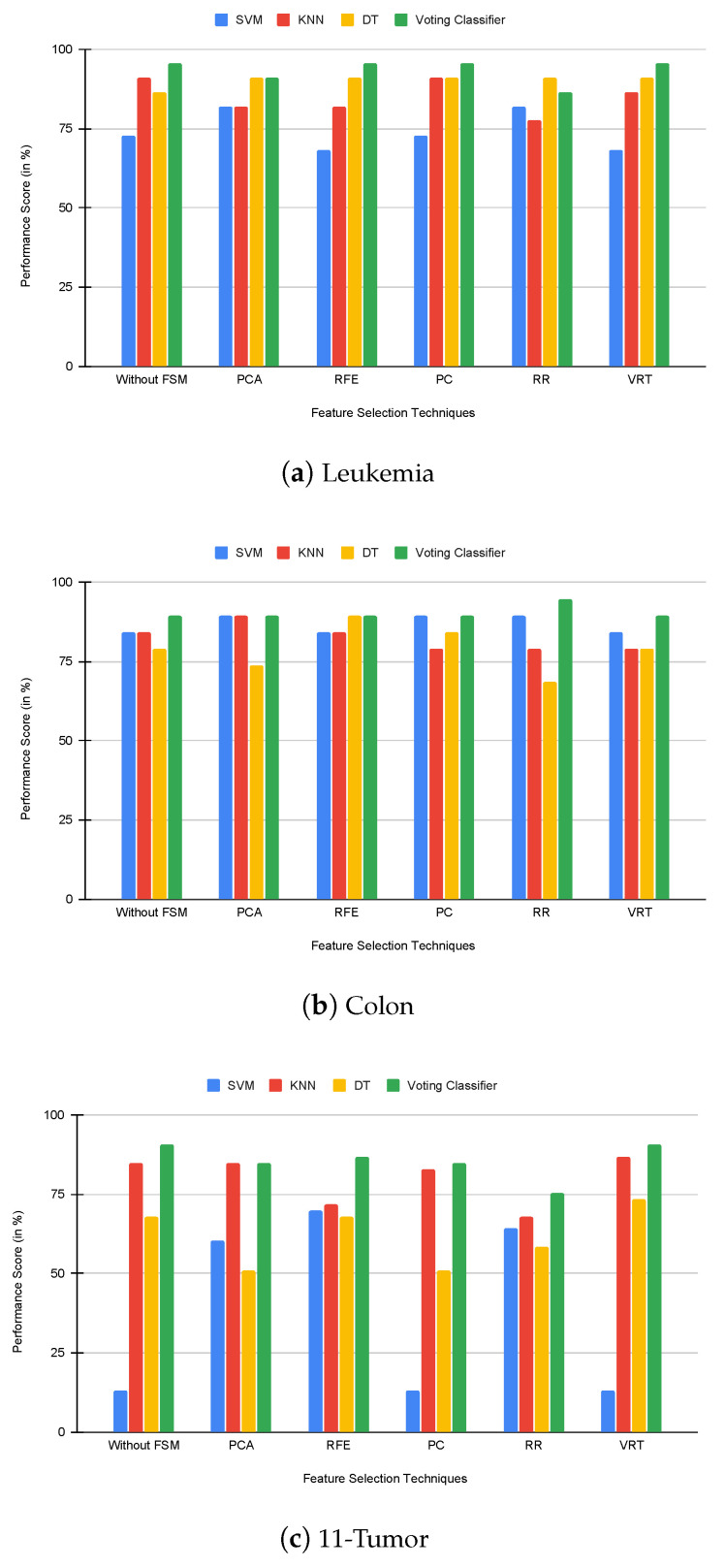
Comparison of FSMs and classifiers using accuracy.

**Figure 4 genes-14-01802-f004:**
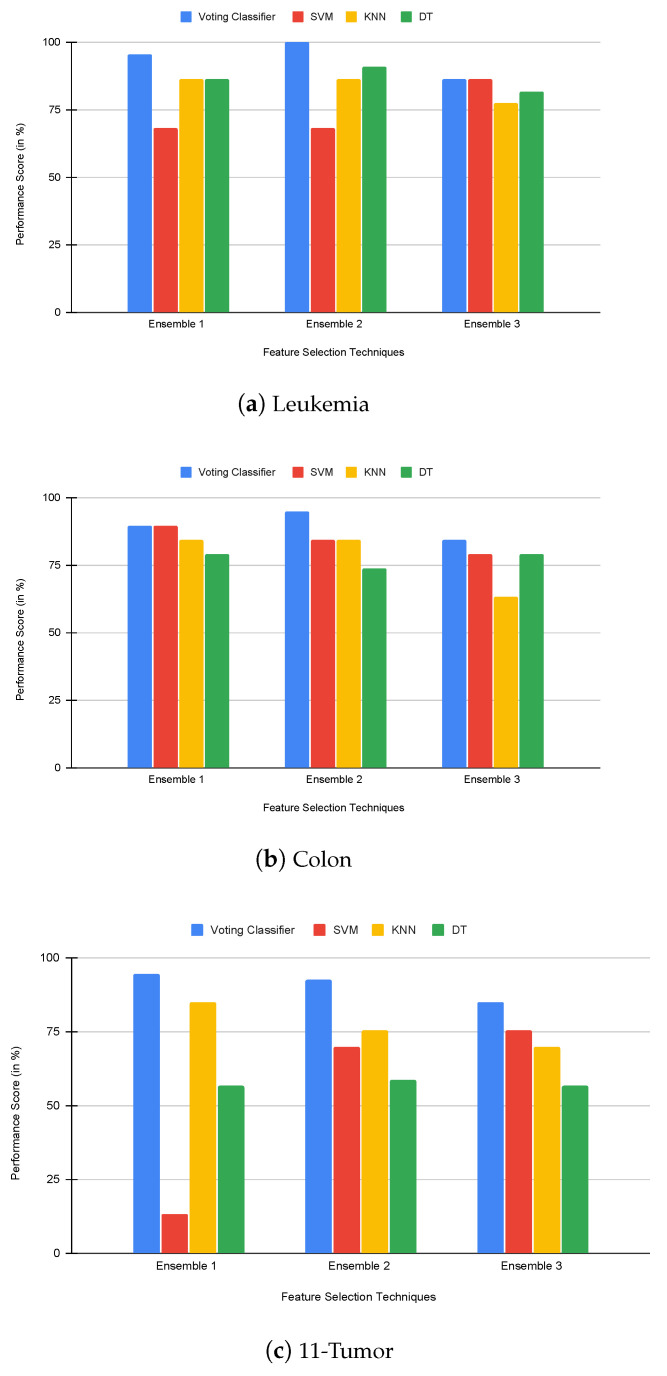
Comparison of FSMs and classifiers using accuracy.

**Figure 5 genes-14-01802-f005:**
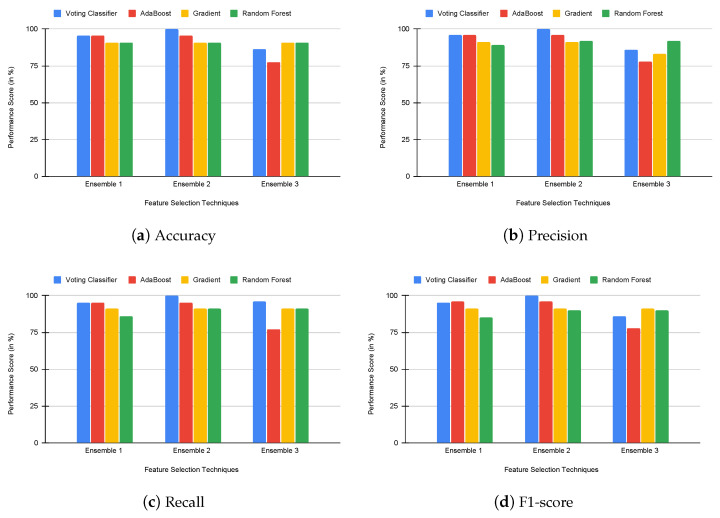
Comparison of voting and built-in ensemble classifiers using accuracy, precision, recall, and f1-score in the leukemia dataset.

**Figure 6 genes-14-01802-f006:**
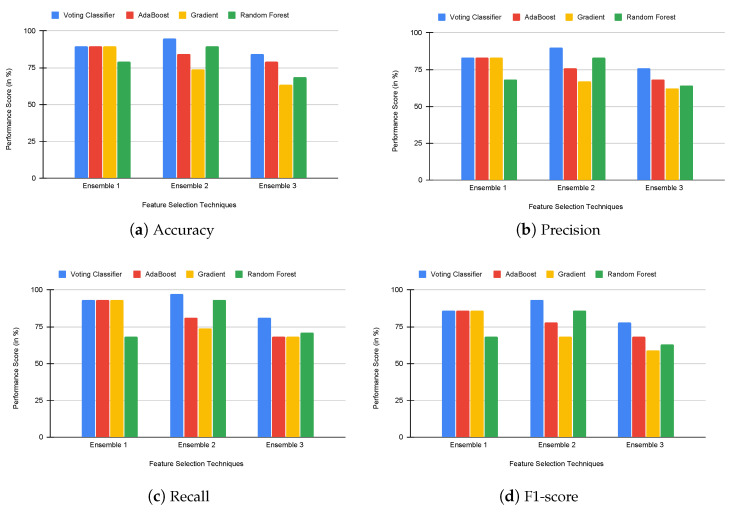
Comparison of voting and built-in ensemble classifiers using accuracy, precision, recall, and f1-score in the colon dataset.

**Figure 7 genes-14-01802-f007:**
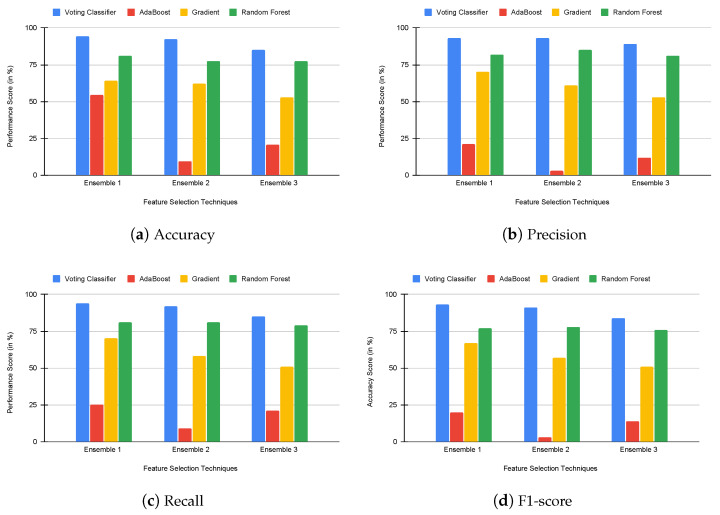
Comparison of voting and built-in ensemble classifiers using accuracy, precision, recall, and f1-score in the 11-tumor dataset.

**Figure 8 genes-14-01802-f008:**
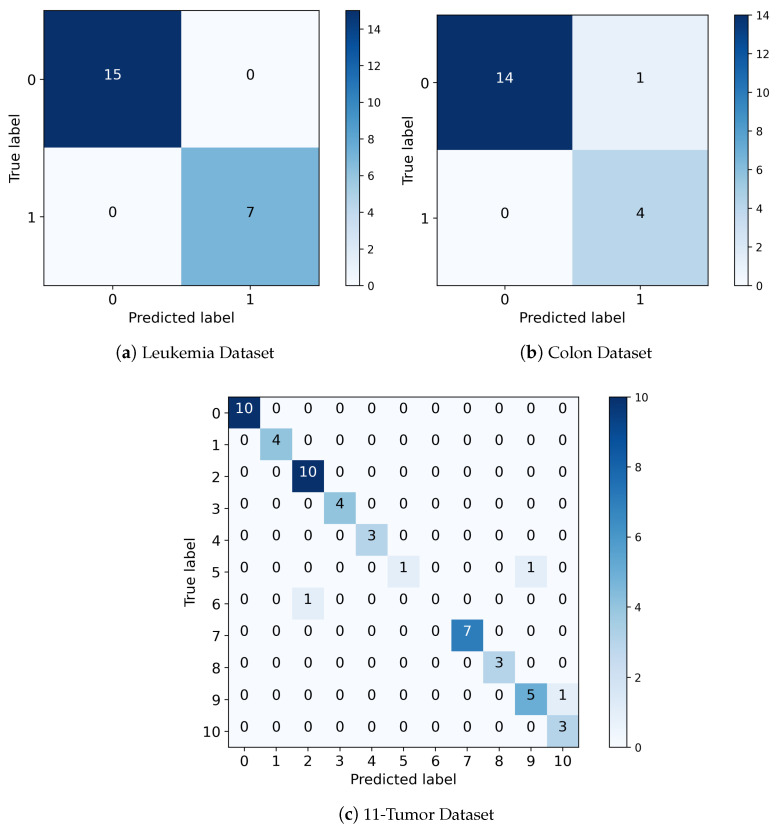
Confusion matrix with best results for different datasets.

**Figure 9 genes-14-01802-f009:**
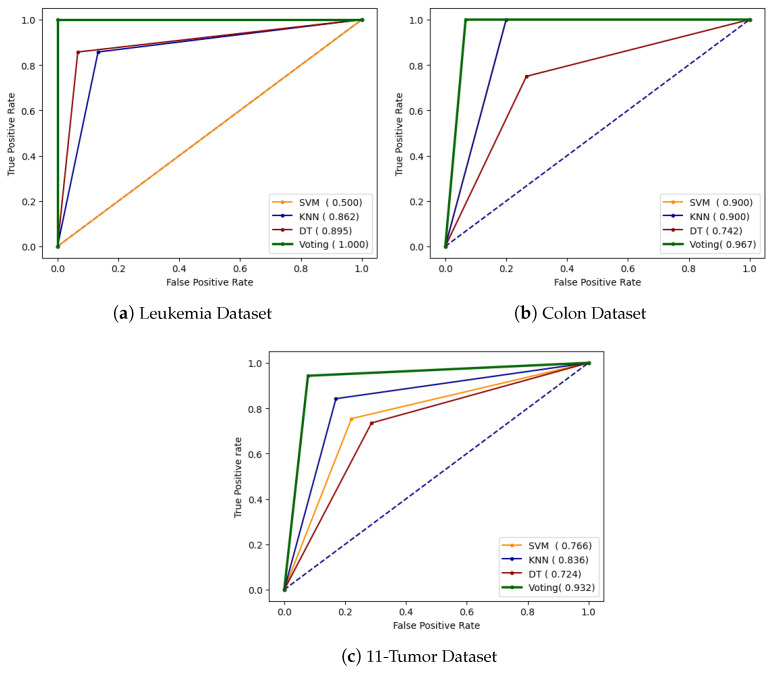
AUROC curve with best results for different datasets.

**Table 1 genes-14-01802-t001:** Summary of related works.

Authors	Dataset	Feature Selection Methods (FSMs)	Classifier	Accuracy Rate
Murad Al-RajabI et al. [10]	Colon	Information Gain, Genetic Algorithm, mRMR	DT, KNN, NB	94.00%
Sung-Bae Cho et al. [12]	leukemia Colon Lymphoma	Information Gain, Euclidean Distance, Mutual Information, Cosine Coefficient, Signal-To-Noise Ratio, Pearson’s and Spearman’s Correlation Coefficients	Multilayer Perceptron KNN SVM	97.1% 93.6% 96.0%
Reinel Tabares Soto et al. [13]	11-tumor dataset	PCA	LR	94.29%
Huijuan Lu et al. [25]	leukemia	MIM-AGA	Support vector machine, Backpropagation neural network, Extreme learning machine, Regularized extreme learning machine	97.62%
AliReza Hajieskanda et al. [45]	STAD LUAD BRC	Grey Wolf Algorithm	DNN	99.37% 99.87% 99.19%
Wei Luo et al. [47]	Lymphoma SRBCT Ovarian	T-test Approach	SVM	100.00%
Ricvan Dana Nindre et al. [52]	Breast Cancer	T-test Approach	SVM	90.00%
Ashok-Kumar Dwived et al. [50]	leukemia	-	ANN	98.00%
Argin Margoosian et al. [53]	tumor	MSVM-RFE	Ensemble-based KNN Ensemble-based NB	85% 94%
Kesav Kancherl et al. [49]	Lung	RFE	SVM	87.50%
Hajar Saoud et al. [51]	Breast Cancer	-	Bayes Network SVM	97.28% 97.28%
Harikumar Rajaguru et al. [48]	Breast Cancer	PCA	DT KNN	91.23% 95.61%

**Table 2 genes-14-01802-t002:** Dataset distribution.

Dataset	Cancer Type	Class	Number of Patients
Leukemia	ALL	0	47
	AML	1	25
11-tumor	Ovary	0	27
	Bladder/Ureter	1	8
	Breast	2	26
	Colorectal	3	23
	Gastroesophageal	4	12
	Kidney	5	11
	Liver	6	7
	Prostate	7	26
	Pancreas	8	6
	Adenocarcinoma	9	14
	Lung Squamous Cell Carcinoma	14	14
Colon	Abnormal	0	40
	Normal	1	22

**Table 3 genes-14-01802-t003:** Selected features.

Feature Selection Method	Number of Selected Features		
	Leukemia Dataset (7132)	11-Tumor Dataset (12,533)	Colon Dataset (2000)
PCA	22	25	24
Recursive feature elimination	5490	5500	1000
Pearson correlation	6991	5611	1011
Ridge regression	3714	6342	1033
Variance threshold	2357	5585	980
Ensemble 1	6580	7528	1354
Ensemble 2	3919	3460	674
Ensemble 3	933	821	139

**Table 4 genes-14-01802-t004:** Comparison of FSMs and classifiers using accuracy.

FSMs	Leukemia Dataset				Colon Dataset				11-Tumor Dataset			
	SVM	KNN	DT	Voting	SVM	KNN	DT	Voting	SVM	KNN	DT	Voting
Without FSM	72.73	90.91	86.36	95.45	84.21	84.21	78.95	89.47	13.21	84.91	67.92	90.57
PCA	81.82	81.82	90.91	90.91	89.47	89.47	73.68	89.47	60.38	84.91	50.94	84.91
RFE	68.18	81.82	90.91	95.45	84.21	84.21	89.47	89.47	69.81	71.69	67.92	86.79
PC	72.73	90.91	90.91	95.45	89.47	78.95	84.21	89.47	13.21	83.02	50.94	84.91
RR	81.82	77.72	90.91	86.36	89.47	78.95	68.42	94.74	64.15	67.92	58.49	75.47
VRT	68.18	86.36	90.91	95.45	84.21	78.95	78.95	89.47	13.21	86.79	73.58	90.57

**Table 5 genes-14-01802-t005:** Result analysis for rank-based ensemble feature selection using accuracy.

	Ensemble-Based Feature Selection								
Classifiers	Leukemia Dataset			Colon Dataset			11-Tumor Dataset		
	E1	E2	E3	E1	E2	E3	E1	E2	E3
Voting	95.45	100	86.36	89.47	94.74	84.21	94.34	92.45	84.91
SVM	68.18	68.18	86.36	89.47	84.21	78.95	13.21	69.81	75.47
KNN	86.36	86.36	77.27	84.21	84.21	63.16	84.91	75.47	69.81
DT	86.36	90.91	81.82	78.95	73.68	78.95	56.6	58.49	56.6

**Table 6 genes-14-01802-t006:** Comparison of proposed model.

Dataset	Reference	Methodology	Accuracy
Leukemia	[50]	Artificial neural network	98%
	[12]	Pearson’s and Spearman’s correlation coefficients, Euclidean distance, cosine coefficient, MI, IG, and signal-to-noise ratio as FSM and multilayer perceptron, KNN, and SVM as classifier	97.1%
	[25]	A hybrid FSM that incorporates mutual information maximization and adaptive genetic algorithm	96.96%
	[63]	Hybrid deep learning based on Laplacian score-CNN	99%
	[43]	Two novel memetic micro-genetic algorithms for feature selection, KNN, SVM, and DT as classifier	96%
	[42]	Soft ensemble feature selection approach based on stacked deep neural network	96.6%
		Proposed methodology	100%
Colon	[12]	Pearson’s and Spearman’s correlation coefficients, Euclidean distance, cosine coefficient, MI, IG, and signal-to-noise ratio as FSM and multilayer perceptron, KNN, and SVM as classifier	93.6%
	[25]	A hybrid FSM that incorporates mutual information maximization and adaptive genetic algorithm	89.09%
	[64]	PCA as FSM and genetic algorithm and ANN as classifier	83.33%
	[29]	IG and standard genetic algorithm as FSM and genetic programming as classifier	85.48%
	[43]	Two novel memetic micro-genetic algorithms and KNN, SVM, and DT as classifier	89%
		Proposed methodology	94.74%
11-Tumor	[13]	PCA as FSM and logistic regression as classifier	94.29%
	[44]	ReliefF algorithm and copula entropy as FSM based on modified gray wolf optimization	89.75%
		Proposed methodology	94.34%

## Data Availability

The selected datasets are sourced from free and open-access sources such as leukemia dataset: https://www.kaggle.com/datasets/crawford/gene-expression?resource=download (accessed on 11 September 2023); 11-tumor dataset: https://github.com/simonorozcoarias/ML_DL_microArrays/blob/master/data11tumors2.csv (accessed on 11 September 2023); and colon dataset: https://www.kaggle.com/datasets/helmy14/colon-cancer-with-gene (accessed on 11 September 2023).

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
