# Peer review of "Cancer Classification Utilizing Voting Classifier with Ensemble Feature Selection Method and Transcriptomic Data"

_genes, 2023, doi:10.3390/genes14091802_

Round 1

Reviewer 1 Report

Authors provide an interesting work and present it in the appropriate way.

There are some limitations that should be addressed in the revision:

1. what is the threshold to define the cancer type

2. I am curious the reason of authors abandoned AUROC in the evaluation, which is the most prevalent matrix to evaluate the predictor performances

3. for table 6, authors used what kind of data to evaluate pervious works? 

Reviewer 2 Report

The paper is very interesting, very well written. The pro problem with used datasets is that they are imabanced  i.e. not equal number of samples per class. Have you considered using oversampling or undersampling methods to improve classification accuracy. If not please cosider it in future research.

Reviewer 3 Report

The authors have presented the work titled as "Cancer Classification Utilising Voting Classifier with Ensemble Feature Selection Method and Transcriptomic Data". The overall presentation seems good. There are few comments which the authors need to address:

1. In several sections such as section 3.2, I will suggest to rename Ensemble because there are a number of tools in biological field with the same name. If possible.

2. Why pearson correlation has been preferred and why not spearmann?

3. Variance threshold needs further expansion and to the point for better clarity.

4. Conclusions seem to be not organised in precise way.

5. Please go through the entire text for some small rewording and rephrasing for english improvement.

Please go through the entire text for some small rewording and rephrasing for english improvement.
